# Canada First Nations Strengths in Community-Based Primary Healthcare

**DOI:** 10.3390/ijerph192013532

**Published:** 2022-10-19

**Authors:** Grace Kyoon Achan, Rachel Eni, Wanda Phillips-Beck, Josée G. Lavoie, Kathi Avery Kinew, Alan Katz

**Affiliations:** 1Education Indigenous Institute of Health and Healing, University of Manitoba, Winnipeg, MB R3E 3P4, Canada; 2Independent Researcher, Victoria, BC V9C 0M1, Canada; 3Department of Community Health Sciences, Max Rady College of Medicine, Rady Faculty of Health Sciences, First Nation Health and Social Secretariat Manitoba, University of Manitoba, Winnipeg, MB R3B 2B3, Canada; 4Department Community Health Sciences, Max Rady College of Medicine, Rady Faculty of Health Sciences, University of Manitoba, Winnipeg, MB R3E 3P5, Canada; 5First Nation Health and Social Secretariat Manitoba, Winnipeg, MB R3B 2B3, Canada; 6Department of Family Medicine and Community Health Sciences, Max Rady College of Medicine, Rady Faculty of Health Sciences, University of Manitoba, Winnipeg, MB R3B 2B3, Canada

**Keywords:** First Nations, community-based primary healthcare, strengths, self-determination

## Abstract

Introduction: First Nation (FN) peoples and communities in Canada are still grappling with the effects of colonization. Health and social inequities result in higher disease burden and significant disparities in healthcare access and responsiveness. For resilience, survival, and self-determination, FN are looking inwards for strengths. This paper reports on the cultural, community, and family strengths that have supported FN communities in developing community-based primary healthcare (CBPHC) strategies to support health and wellbeing. Methods: The study was a partnership between university-based researchers; The First Nations Health and Social Secretariat of Manitoba; and eight First Nation communities in Manitoba. Community-based participatory research methods were used to engage the participating communities. One hundred and eighty-three in-depth, semi-structured key informant interviews were completed between 2014 and 2016 with key members of the First Nation communities, i.e., community-based health providers and users of primary healthcare services, representing all age and genders. Data-collection and analysis were conducted following iterative grounded theory analysis. Results: Community-based healthcare models based on local strengths support easier access and shorter wait times for care and compassionate care delivery. Resources such as homecare and medical transportation are helpful. Community cooperation, youth power, responsive leadership, and economic development as well as a strong cultural and spiritual base are key strengths supporting health and social wellbeing. Conclusions: Locally led, self-determined care adds strength in FN communities, and is poised to create long-lasting primary healthcare transformation.

## 1. Introduction

Difficulties faced by Indigenous peoples in Canada are well-documented and understood [1,2,3,4,5]. Indigenous Canadians, i.e., First Nations (FN), Inuit, and Metis continue to show poorer health outcomes than the general population [6,7]. Many current challenges result from historical mistreatment through lengthy colonization, successive draconian policies and practices [3,8,9,10,11], targeted oppression [8,12], and ongoing neglect [13,14]. FN health across Canada is an area of particular concern. It presents worrying trends and disparities indicating poorer health outcomes compared with the general population [15,16,17,18,19].

For Indigenous Canadians, self-determination has been essential to cultural continuity, ensuring community wellness and improved health outcomes [9,20,21]. However, self-determination has been pruned back to a genre of self-administration constrained by limiting access to basic resources and healthcare services. Generally, primary healthcare for First Nations communities in Manitoba is funded and delivered via 21 federal and 3 provincially funded nursing stations, along with funding for community-based ancillary services. In the remaining 38 communities, the federal government transfers funding directly to FN communities for the delivery of limited public health and community-based services, maintaining it has no mandate to fund primary healthcare in these 38 healthcare facilities due to their proximity to provincially funded primary care in the surrounding areas. Gaps abound in services to Indigenous healthcare and inadequate primary healthcare for FN in the Canadian context [22,23].

FN peoples living in rural and/or remote communities in Manitoba travel from 2 to 15 h to access healthcare in facilities located in cities. Some take two airplanes or 10 h bus rides to access healthcare services. Emergencies are transferred out of the community often by air evacuation at the approval of a physician. FN health is also undermined by overcrowding in homes, contaminants in the environment, poorer living conditions, food insecurities, low education and literacy rates, lack of or unstable employment stress, and racism [7,24]. Colonization has left psychological impacts of trauma and threat of disconnection from traditional knowledge and the nourishing benefits of the land [25,26]. It is critical that transformational change comes from FN peoples utilizing community-based healthcare. They are the ones who experience, firsthand, limitations of differential models of care [23,27,28] and hardships associated with traveling long distances for care. Despite existing difficulties and the fact that they tend to be highlighted and discussed as deficits, it is important to draw attention to the remarkable strengths and solutions FN are leveraging in the face of unique difficulties.

This study was carried out in response to Manitoba First Nation calls for further research into individual community-based strengths (e.g., roles, responsibilities, participation, resources, and perspectives) in local primary healthcare. The study sought to understand existing strengths in FN communities supporting CBPHC despite the myriad challenges faced in administering primary healthcare with limited resources. This step was premised on the understanding that supporting already-existing strengths may be the determining factor to transform CBPHC in FN communities and build stronger communities. The (IPHIT) project is a 5-year research project funded by Canadian Institutes of Health Research (CIHR). It is a primary partnership among researchers at the University of Manitoba, the First Nations Health and Social Secretariat of Manitoba (FNHSSM), and eight FN communities in Manitoba.

### Background

Various concepts are applicable to the study of First Nation health and community-based creation of healthcare programming, stemming from decolonizing and Indigenous research. For instance, cultural colonialism has been analyzed as an extension of colonial state power through cultural knowledge, biomedical activities, and health institutions, as well as the systemic subordination of Indigenous conceptual frameworks and cultural identities. New knowledge amasses on global inequities and the cultural dimensions of present-day struggles for self-determination as decolonizing discourses critique political and socioeconomic systems [29,30,31,32,33,34].

Reconnecting the severed pieces is paramount to a decolonizing and Indigenous approach to health and healthcare. When the Europeans came to North America, they seized the land based on the principle, terra nullius, meaning “nobody’s land”, thus justifying the physical dislocation of Indigenous peoples from their land and resources and the state’s occupation of them. Ongoing resource extraction and violence against women perpetuates the negativity, creating conflicting, complex environments upon which current health discussions exist. Disconnection from previous harmony and deep spiritual connection to the land negatively impacted First Nation health. Colonization disconnected Indigenous peoples from traditional ceremonies, traditional medicines, traditional land uses [35], and their worldview [36]. Family and individual relationships as well as those with the inanimate world were also broken in the process [37]. Community journeys from the cycle of violence to the circle of wellness have been conceived of as ‘relational determinants of health’ [38].

A great deal of research on the intergenerational impacts of the residential schools reveals the incomprehensible level of violence that Canadian Indigenous peoples have been enduring [39,40,41] and its impacts on health through generations [42]. Holding the silence [43,44] and the ongoing denial of the atrocities on the part of Canadians [45] have further eroded health. Recently, more than 1300 unmarked graves were uncovered across sites of five residential schools [46], news that validates something within Canada’s history of which Indigenous peoples were already well aware.

Self-governance and self-determination are essential aspects of community wellness outlined in health legislation. An inherent right of self-government is recognized by Canada and protected by Section 35 of the Constitution Act, 1982. Despite such recognition, there is a distinct absence of Canadian public policy supporting Indigenous health at both national and provincial levels [47]. Historical processes and values inherent in the connection between self-determination and governance to health and healthcare is discussed in the literature [48]. Self-governance is a critical component of community healing and in repairing ruptures and discontinuity of transmission of traditional knowledge and values in asserting collective identities and power [49].

Levi Straus’s conceptualization of symbolic association in health and healing [50,51], offers a way of looking into the meanings that people ascribe to methods in healthcare, whether these are traditional or biomedically oriented. Symbolic efficacy of Indigenous healing practices regards the influence that beliefs, myths, ancestral stories, etc., have on getting well—physically, mentally, and spiritually.

Self-care is defined as a naturalistic decision-making process addressing both prevention and management of chronic illness with core elements of personal maintenance, monitoring, and management [52]. Self-care constitutes a basic structure that micro-groups generate in order to live and survive, but that biomedicine has co-opted as a policy of its own making, repositioning the locus of power from individuals to the medical institution. Within First Nation communities, self-care encompasses family and community care. Homecare falls within this theme. Self-care, (including care of family and community) was interfered with by legislation prohibiting cultural practices, i.e., prohibitions against community childbirth, hunting and trapping, use of traditional medicines, and participation in ceremonies, to name a few. Recent upsurge in food sovereignty and community birthing programs combine self-determination and self-care in the revival of health at community levels [53]. Strengths in healthcare planning include community practices inside and surrounding both traditional and biomedical paradigms. Indigenous worldviews are encompassing of determinants of health that span time and space, connecting individuals to complex ecologies and ancestries and to dimensions of spirituality not considered in biomedicine [54].

Community involvement is a central tenant of First Nation healthcare programming. The creation of space within development and implementation of healthcare for the community honors their voices and experiences, cultures, and traditions [55].

## 2. Methods

Methodologically, grounded theory was coupled with and based in participatory research. Grounded theory offered an iterative study design, theoretical (purposive) sampling, and system of analysis. Data collection and analysis were generated simultaneously in partnership following community-based leadership at all points of the research process. At each stage of the data collection process, analysis—involving constant checking by community research assistants—informed the next cycle of data collection. Community-based research, conducted in participation with the First Nations and following their lead, is based in self-determination, decolonizing, and Indigenous research as well as in principles of collaboration, egalitarianism, and social equity. Ultimately, this process allowed for a blending of scientific inquiry with social action to create knowledge that is relevant and useful to the First Nations in facilitating transformative primary healthcare in their communities.

This community-based participatory research (CBPR) was guided by a grounded theory approach [56,57,58] All partners were engaged in collaboratively designing and implementing the study. CBPR calls for authentic and meaningful community engagement [59,60] to generate actionable data. It is widely used in Indigenous environments [59,61,62]. Grounded theory is well-suited to guide the CBPR process because it allows the data to lead the way into information exploration, arriving at themes and theory from the majority’s perspectives. We assumed an open approach throughout the study and learned from the data, which were being analyzed as they became available. This iterative approach allowed us to map out themes where theoretical saturation had been achieved and to identify gaps. This then guided our sampling process for further interviews.

In addition to ensuring all emerging themes had been thoroughly explored, we also compared data from participating communities as data were being gathered. Detailed descriptions of methods and engagement processes have already been published [30,63,64]. We used a semi-structured, conversational style, and we encouraged storied responses in which respondents were free to engage the subject matter expansively [65,66]. Interviews were completed between April 2014 and December 2016. Following all interviews and preliminary data analysis, initial results were presented in all participating communities so that respondents and other community members could inform the interpretation of data from their community. All communities came together to review and validate the data through data validation workshops continuing into 2018. An FN member is an individual who is recognized as belonging to a First Nation, defined either by the community or band itself or by the Indian Act.

Ethics approvals were obtained from the University of Manitoba Health Research Ethics Board, Ethics approval # HS17012 (H2013:454) and First Nations Health Information Research Governance Committee (HIRGC). FN Chiefs of Manitoba mandated the HIRGC since 1998 to oversee research ethics processes on behalf of Manitoba FNs. Written agreements were signed between FN communities and researchers in keeping with HIRGC requirements including FN Ownership, Control, Access, and Possession (OCAP) principles [67]. Following both approvals, data collectors obtained informed consent from respective communities and individual respondents prior to conducting interviews. Researchers adhered to FN ethical standards, which require them to ensure that the study and all data collected would be of direct benefit to FN communities, as much as possible. OCAP and other principles are intended to regulate and guide research that involves FNs, so research is conducted respectfully and meaningfully for community partners. It is imperative that research data become a tool to strengthen FN inherent rights to self-determination.

Participants were individuals from FN communities representing all geographical locations in Manitoba including isolated, semi-isolated, and non-isolated communities. They represented four of five FN languages spoken in Manitoba: Dakota, Dene, Cree, and Ojibwe. Advisory committees in each community oversaw the research process and provided ongoing feedback to researchers with an aim of ensuring respectful adherence to necessary protocols, and that information would be readily made available to communities upon request.

Purposive sampling targeted community members, Elders, i.e., health service users, and FN and non-FN healthcare workers in the community. Individuals were selected based on their knowledge or experience with community-based healthcare systems or with traveling off reserves for care in towns and cities. For some, personal experiences translated into leadership positions in which they worked to make changes to the existing systems. The research team recruited and trained local research assistants (LRA) to collect data. One hundred and eighty-three interviews were completed with individuals from eight communities. Each LRA translated questions and responses into respective local languages as necessary to accommodate non-English speaking participants. All interviews were transcribed verbatim and sent back to communities for validation. LRA reviewed, translated, and explained terms, meanings, and contexts for increased clarity throughout data analysis.

Preliminary data analysis was conducted using an open axial coding system to identify key ideas and themes, which was then uploaded to the NVivo 10™ software (QRS International, Kitchener, ON, Canada) for further thematic coding and analysis. LRA and community members in all communities were involved in interpreting preliminary findings in their data through community data presentation sessions at which the entire community, including local leadership, was welcome to listen and provide feedback on the results. This process validated the data and significantly reduced the ‘researcher effect’ on the analysis [68]. It also facilitated data dissemination and implementation of efficacious findings by the communities as is being increasingly recommended by funding bodies [69]. The results summarized below emerged from interview data and from interpretive discussions during data validation sessions. Quotes have been selected to represent themes emanating from all participating communities and that were common to majority of respondents. (COREQ Checklist is available in the Appendix A).

## 3. Findings

Participants highlighted strengths supporting health and community-based primary healthcare delivery. Nine inter-connected central themes are highlighted. These are: (1) having a cooperative and engaged community; (2) committed and responsive leadership to oversee and inspire healthcare sectors; (3) activities that support social connection and wellbeing in the communities; (4) ensuring patients have positive experiences with an accessible healthcare system, including provision of timely care, medical transportation, and compassionate care; (5) homecare for Elders and others who may be immobile; (6) ensuring a connection with the land; (7) spirituality; (8) culture; and (9) economic development to address the social determinants of health (SDH (Figure 1)).

### 3.1. Cooperative Community

Community members were seen as a strength, more so when community is engaged and cooperates in making life easier for each other. Individual attributes, strengths, experiences, and opinions were seen as contributors to community wellness. Everyone has a role; everyone has something to offer. When people are engaged and working together towards common goals, the whole community is seen as moving forward in a healthy way. Of the themes, cooperative community was given greatest importance by the participants.

I think that the strengths of the community reside within the people themselves. There is this thing that they have—resilience. There is a resilience that people have in this community that no matter the circumstances they get back on their feet and they don’t let life’s circumstances take them way down…—I see that as a strength. And they see support for each other when the need is there.(F006)

I guess everybody sees this like our community is different from others. It seems that in our community there is always people that are willing to help each other. We have an abundance of help that comes from people. Like say if there’s a death, everybody wants to help. If somebody is in trouble, accidents, people come. People come together and they help.(C007)

Respondents identified having a younger population in the communities as well and spoke of the bi-directional learning that occurs between each end of the generational spectrum.

[Our] community has a high population of youth and I think having the youth in the community that is a strength…we’re actually learning from the youth. And it goes hand in hand. Youth learn from Elders and Elders learn from youth. And by Elders and youth working together in the community that’s a strength. You know, coming together and sharing their knowledge and providing that support to one another. I find that’s a big strength.(D005)

Particular strengths of each generation were reported as essential to self-determination and to the creation of wellbeing. Intergenerational dialoguing was discussed as a subtheme of cooperative community indicating an essential relationship of individual involvement in connection to others in the group.

### 3.2. Leadership

Respondents attributed good community-based healthcare with responsive leadership that understands the communities’ needs and tries to meet them.

What I found is that the health leadership is trying very hard to help our First Nation. That’s what I find unique about our community. They’re trying to help First Nations [people] from 0 to 100 years old, in any possible way. The services are there; people aren’t scared to use them. That’s what I find unique.(B017)

This quote captures a widely shared sentiment appreciation, that FN leadership is responsive and approachable in the communities. Within the leadership theme, it became apparent that leaders were community members affected in the same way as other community members by colonial practices, etc. They were part of the circle of community membership but with the added responsibility of articulating and addressing community interests. Institutional hierarchies were not a part of the community-based leadership dynamic. Instead, community members, health administrators, providers, and the leadership lived and worked together within a circle of health.

### 3.3. Activities in the Community

Respondents appreciated that nature provides free opportunities for leisure and recreation. There was a healing element to connection to the land.

You know, it’s unique to have a community like that that’s got lots of land, land to use, lots of water to use. I find that unique because a lot of [other] communities don’t have that. I lived [off reserve] for a long time there and a lot of people look for a place to go swim over there in other places. Like they look for places to go camping but there’s lots of rules, you know, when you go camping. But here you can go anywhere, anywhere in the bush here to go camp. You know, it’s unique like that that we can use the land pretty much and not ask anyone to use the land. We don’t have to go ask for a permit or things like that. I see that as unique and it’s good to have that.(C011)

Another respondent echoed others in describing a thriving community with activities taking place, bringing people together for learning and socialization.

Lots of events always taking place… this winter they were showing them [kids] ice fishing, how they hunt and trapping, how to make traps and …I know my uncle said that they asked him to do some teachings. Like that’s another thing too is he said he goes down there, and they gather, and some families want to know their history. So, he sits there and tells them who they’re related to and tells them their history. That’s another thing they do down there, it’s kind of like a gathering place for different events like I said…a community-gathering place for our people. And everybody camps and it’s really good and it’s growing. I know they have more things that are being added on. Like the big event they have is the country days, big country singers they bring out and that’s good. And then they have our radio station which is another good thing and that probably will even get bigger. Maybe they’ll even get a television station here. We have a lot of good things, I’m proud.(D002)

Our traditions are still alive. There’s still caribou hunting, they are still dry meat making, there’s still people fishing. Beadwork, you still see it. And we have a sense of pride when we see that. Even our language, when we talk and nobody else understands and there’s two of us speaking our language, there’s pride in that. That is a good strength.(FFG1-1)

These activities that bring people together were said to elicit a sense of pride in members and of the culture and community to which they belong. Natural environments, traditions, land-based activities, connection, relaxation, and stress reduction were subthemes of activities in the community.

### 3.4. Positive Experiences with Community-Based Healthcare System

#### 3.4.1. Accessibility—Medical Transportation

Medical transportation, where available, was valued, especially by those who did not own vehicles or have public transportation options. These people may not otherwise be able to access healthcare facilities within or outside the communities.

The only difference [with off-reserve] would be that the medical, we have medical transportation, like, even for the young. I guess off the reserve just the old folks have it I guess. We got it for everybody. Even for guys that have vehicles, they still get on the bus, don’t they?(E009)

A lot of things are getting to be a better… now I’m older they come to my home, which is just nice, instead of us going to the nursing station. Then, at that time, we didn’t have no transportation to go to the nursing station, and we used to walk. Now we have a medical van transporting us to and from.(GFG004-3)

Providing rides for people that [don’t have any] so they can come see the doctor or the nurse or the dentist or anything is a good idea. And maybe increasing the number of vans so people can access our resources would be wonderful.(H005)

It was said that medical transportation increases access to care for people who would otherwise not be able to travel to appointments. Community-based healthcare providers also assisted community members by driving them to and from appointments, where possible, in order to try to enhance resource efficacy.

#### 3.4.2. Coordinated Care

Respondents recognized that wait times in health facilities were shorter in communities compared with healthcare facilities off reserve.

Well, I mean I think we’re lucky I would say because if you go to an emergency in Winnipeg how long do you sit there and this is nothing compared to being in the city. Like you have a nurse here or even have a doctor here sometimes, but I think we’re lucky. It’s just that what I said before about the nurses, they have more compassion for their job.(C014)

I would think just faster service… I would say on the reserve you get pretty good service too, yeah. Anytime I needed help or my daughter or any of my kids needed something, I phone here, they set it up, they let me know “Okay, we got you in this time for this time,” and all that. I don’t got to worry about it but off the reserve I would be worried. Oh, I appreciate it.(E006)

Shorter wait times can result in patients being more motivated to attend appointments, conditions may be detected early, and treatment commenced in a timely manner. Study participants said that having healthcare providers who were also community members allowed for a more seamless delivery of services. Community-based healthcare services became more accessible, more a part of everyday life on the reserves. Physical as well as personal and cultural proximity were subthemes. This theme also connects to the theme discussed above regarding leadership and cooperative community.

#### 3.4.3. Compassionate Care

Respondents said it was considered ‘a strength’ when they have positive experiences at healthcare facilities with local workers in the community. This, they said, occurs when workers can relate to their experiences and were compassionate, helpful, and efficient.

We have some really good workers here that are dedicated to their jobs, and they try to help as best they could to help take care of the person at the time of need, and refer on, if need be.(E016)

The care provided was reported to be more personal and compassionate when providers know and build relationships with the patients.

I think maybe that’s the difference. And they’re more, cause I think maybe too they know you here, they’re more, they have that caring, you know, caring part. But I think community D is actually up there, like they’re careful.(D002)

There again, they say we have to have certificates of all kinds in order to provide the good care we need. I think we have our own resources here for people-, caring people. I’ve seen one family provide palliative care very good to a relative of mine. Some people feel better with community people.(E010)

Again, the notion that self-care is an important determinant of health, one that comes from community and is outside of biomedicine, is evident within this theme. Community members provide compassionate care to one another—to family members. Health resources are necessary, and their availability provide families with the means to better support one another. However, the know-how and the compassion come from tradition, passed down through generations of caring.

#### 3.4.4. Homecare

Respondents said that having homecare, especially for Elders, is a definite strength. This may also be because the alternative has been to remove frail Elders and patients from the community to healthcare facilities in distant places away from family and the communities.

Well, the services I guess would be, like if there had to be a difference to off reserve people, is probably homecare because we don’t offer homecare to off reserve provincial.(B003)

Well, I think that homecare is important for the benefit of the Elders and residents who need healthcare at home and get healthcare. It’s something that’s really important.(D003)

I think on a couple of occasions already I’ve had personal service provided to me from the health centre when I had an operation. I had a nurse come in and change the dressing on my surgery wound, also when I had an operation on my chest the second time. I had my foot and part of my leg amputated. I was looked after by the health nurse.(H020)

Homecare and healthcare connect formal with informal health resources. Biomedical and traditional care join hands to offer community members, and in particular, the Eldest of the membership, the most compassionate care.

#### 3.4.5. Connection to the Land

Respondents spoke about having a sense of connection to the land, which, in turn, supports a sense of wellbeing.

That would be it there, I think. More into the land, it’s the land that binds us. It’s the land that defines the Native person; it’s in our blood. Like I said, I don’t understand. I tried to look at it and say, “Why is this? Why is that?” I’ve missed so much growing up-, not really growing up here. The old people have come and gone, some of the old people I have talked… you sense that, you sense the attachment to the land. I raised my kids on the trap line, I taught them from the trap line. My Dad even told me that’s how we began. He was raised at the trap line. Blood was in the land and land was in his blood.(A007)

Others spoke about taking care of their immediate environments to promote wellbeing and sourcing healthy nutrition from the land as well.

Nice looking lawns, families helping families, in general, like a good family relationship, a good healthy family relationship.(B017)

We’re proud of getting gardens and trying to go back to the old ways where it was—I know it was harder but it was actually better…I think too about the kids, they’re getting so obese you know and it’s because we wanted to make life easier like you know but easier ain’t always best.(CFG2-4)

Natural foods and I know when you think of the Elders, they kind of live like a long life because they lived off the land and they worked really hard and they had a lot of physical activity. Because like myself as a little girl, we went to pick berries, you know, according to the seasons. We still pick the berries. And I think like you’re canning, you’re preserving and there’s a lot of … there’s that cultural part of a way of life. Yeah and … whereas today we’re at a stage where we want to embrace that cultural life again because we know it was a good life and it was a healthy way of living within the community. That’s the best way to explain it but you know what I mean.(D005)

I think the uniqueness of the community here is the land we live in and how we use our environment…I believe your surroundings around you have a lot to do with the way you live and your health. [The] environment you put yourself in or how you take care of your environment. I believe that’s a big thing, keeping it clean, and healthy as well.(E016)

Natural resources, food, and coming back to the land were seen as major and defining First Nation determinants of health. The decolonizing element within particular discussions is clearly revealed in the return to tradition and Indigenous identity via reconnecting with the land.

#### 3.4.6. Spirituality

Resurgence of traditional spiritual practice was seen as strength, offering options for people to receive spiritual care when needed without fear or stigmatization.

They have these people that seek traditional healing and traditional help and they know the history of it. Before I came into this community, it was more focused on Christians, but I can see now there is more open [traditional practice] now, they’ve accepted the people. They’ve begun to accept it and it’s starting to be open to people. The people who carry the traditional healing and light, they’re out there. You can find them if you need something from them. You know who to go to get advice but maybe 20, 30 years ago you weren’t, you couldn’t, you know? That’s something positive to me is that we have options. Yeah, their strength is that when the people seek for help, they’re able to get it; we have different options. We can see children following that too. They want to learn how to dance and how to make drums and that.(A014)

Elder roles in maintaining spiritual practice in FN communities were also highlighted. One respondent referred to this practice as “the red road” that seems to be facilitated by “Spiritual Elders”.

In this community, there is currently seven healing and sweat lodges present. And I think all the community members who follow this way of life would you know, live a balanced life. We call it the Red Road. Spiritual Elders take your inner child and work their way out toward an adult, to bring everything out that has happened in the past, so you can live a balanced life, whether it was abuse or alcohol or there’s a number of abuses that could be said. That’s my strength is when I go into a sweat lodge, is the same thing as a healing lodge for me. The Red Road… it’s you know trying to live that balanced life, the seven teachings that the creator had given us to follow. The [FN] communities that I’ve gone to, there is a healing lodge that exists in the community and there’s always a Spiritual Elder that comes from that community. And I guess that’s the strength of our communities.(B021)

Respondents discussed spirituality in terms of an ongoing process of exploration as well as in terms of something that was always there, a home, a connection to self, community, land, and spirit. It was the meaningful something that always belonged to them but that they didn’t always remembers was there. Spirituality is a strength for community members, which like the other themes, has survived colonialism.

#### 3.4.7. Culture

Respondents identified family values of respect, sharing, helpfulness, role of women, language, and community as key strengths utilized in the community and in workplaces.

Yeah, that would be one here because I know even when I had my own experience with a funeral, I had relatives come out of town. And they couldn’t believe how everybody helped out and how they have the and how they did that, and you know, and how they thought of that, and how everybody come and shook everybody’s hand, they thought that was really different. They’d never seen that. They liked that.(A008)

Workers here all practice their culture. They dance, they sing and [one of them] was a sweat lodge keeper. We have a lot of women that work in here… They’re all grandmothers and mothers and they all still speak their Ojibway language, which is an asset both to work with the older people and for the younger people. [Our] language is still here and I think that’s unique in our community because I know our younger generation don’t speak the language but all our older generations speak the language. They’re speaking to the community so we still have that in our health center and I think that makes us unique.(B005)

Our community has traditions. By traditions, I mean things that people did together. People, you know, fish together, the helping of one another; you need people to help each other. People help with raising their children. I know my grandmother and her sisters were like the people that could correct you. My dad’s sisters, my aunts, they could correct you and my dad wouldn’t say nothing, my mom wouldn’t say nothing. [They’d say] good, it’s good you corrected my children; go ahead, because if they’re doing something they shouldn’t be doing then good, you correct them. People took their kids to church in the morning, took them in the afternoon, whenever they went and it was good. The kid knew, the child knew to respect people and also how to pray. Also, when I’d see an old person like we’d pass an older person and my grandma would say, “Say hi to your grandma” and I’d look confused at first, then I got the hang of it. Oh, yes, okay. Every old person you met, whether it was a man or a woman, if it was a woman that was your grandma and grandpa the other one, you know? You have to call them that. You couldn’t call them by their name and same with your parents, you wouldn’t call them by their name. We were taught to respect all Elders in the community and then in the community it was fun. You’d meet somebody, you said hi, you talked to them and then my grandmother and I would stand there and never get tired but I knew I’d have to stand there while she chatted with somebody, you know? And stuff like that. That’s what made it unique for me, unique community.(C010)

Availability and use of traditional medicines was pinpointed as a strength in the communities.

The strength, I think, each different community has their own strengths within the people. We recently started a culture camp, which is a good thing to be starting, things where people could learn. Strengths, here there’s traditional medicines in the community that are being used and utilizing the western medications, as well.(E016)

#### 3.4.8. Economic Development

Respondents indicated that economic activities in the community that create opportunities for members are a strength. This is important since some are unable to secure jobs outside the community. Economic development included the subthemes: community active involvement in health enhancing activities, cooperation, and moving forward. In addition, mention of arts and crafts referred to cultural and traditional activities within economic development.

Some of us are good at stuff you know what I mean. They just don’t give us a chance. I had a hard time finding a job in Winnipeg.(CFG102)

Right now I think it is work… a lot of jobs being provided… And our strength, our health is getting better. I think health is getting better. I see more guys working more people don’t have to leave the reserve for that. …Lots of housing… I’m kind of proud of that, that we’re doing this, doing it for ourselves this time around. For the first time, I find it unique right now is that we’re doing it ourselves, showing we can do it ourselves.(E006)

They’ve got the sewing program and they do arts and crafts, and they have that case displayed out there for them to sell crafts, I really like that.(GFG004-4)

## 4. Discussion

This paper focused on strengths within healthcare emanating from First Nation communities. Together, themes spoke to community interrelationships, active engagement in health and health activities, leadership and cooperation, accessible services, specificity of services, i.e., homecare and its focus on Elders, and, particularly, for the last 4 themes, tradition and interconnectivity with the land and its healing resources. The themes highlight the strengths of each generation, e.g., youth may bring fresh ideas and energy into activities, while Elders pass on traditional knowledge about the land, sacredness of spiritual ceremonies, and so on, as well as importance of intergenerational connection. The nine themes depicted above are essential to self-determination and active engagement in healthcare provision and to creation of community wellness. Here, community members are not merely ‘recipients’ of healthcare, rather they are promoting the development of healthcare and an overall wellness fitting with their needs and interests.

The first two themes speak to cooperation between members and their leadership. In terms of health governance, the provision of health services is dependent upon the interrelatedness of roles and responsibilities of community members and leadership to one another and to the tasks at hand. Strong and responsive leadership brings the interests and cooperation of the people to the forefront (first theme). The work of the leadership is to engage with and listen to the membership, understand their needs and interests, and to work toward their realization. Community participation is as essential as leadership mobilization; one cannot be implemented as community-driven without the other. As we witness by the fourth theme, access to health services is key. Accessibility in terms of transportation and distance to the services, timely service, and compassionate care are unique emblems in community-based care. The final four themes of connection to the land, spirituality, culture, and economic development, relate to a fully encompassing, specifically First Nation healthcare system throughout the lifespan, taking into consideration the social determinants of health. Spirituality and culture are highlighted and underpin a First Nation conception of what it means to be whole, healthy, and to have a meaningful and engaged life, underpinning all aspects of living, i.e., including economic development, education, and social activities in communities.

Strategies to address existing inequities include developing integrated measures to support community-oriented primary care and proactive equity considerations that do not superimpose solutions, promoting existing strengths and focus on the entire population or community [70]. The call to elicit and support community strengths is particularly welcome and long overdue. First Nation communities are taking the lead on implementing strength-based, culturally embedded approaches to address historical and current structural realities that limit health and healthcare [71,72]. Self-determining approaches are intended to better serve FN peoples and communities in ways that heal, promote long-term wellbeing, and can be sustained by communities [72,73]. There is wide recognition by FN in this study, of the importance of building striving communities that, in turn, nurture health for all citizens; therefore, community is seen a strength, as is the role of strong FN leadership built on Indigenous values [74]. Support of already existing strengths and community-led processes and innovations has been advocated as crucial for local buy-in when aiming for improved sense of health and community-based healthcare transformation.

Community-based primary healthcare models that facilitate holistic Indigenous approaches, have been shown to more comprehensively address varied determinants of health and wellbeing than mainstream systems, which tend more toward addressing specific concerns [23]. In practice, such models may mean health programs operate collaboratively, encourage relationship-based service provision in which healthcare workers in communities are aware of client histories and experiences, and thus, strive to be compassionate [75]. This is premised on evidence that community-based workers with long term commitments who have developed appropriate trust within the community may also be better placed to manage quality improvement and surveillance measures in order to understand disease and healthcare trends, both of which have been recommended as possible ways to improve care for Indigenous peoples [76]. Such models create positive healthcare experiences for patients. This process is supported by strong and development-oriented leadership that understands and works to meets the needs of the community [74].

Wait times to see a healthcare provider, which may or may not result in medical treatment, is reported by respondents in this study to be shorter in FN communities. This may be attributed to community-based healthcare models that are essentially patient-oriented. To the extent that they result in treatment or referrals, they can be a huge factor enabling better medical outcomes than those that FN patients tend to achieve outside of communities [77]. Medical transportation, where available and reliable, is flagged as a key healthcare access facilitator within and outside FN communities, as was homecare when sick individuals or people living with disabilities are unable to, or choose to, remain in their respective communities.

Extensive and rapid pace engagement with contemporary advancements is being balanced with traditional land-based activities to foster supportive relationships among people and with the environment [78,79]. Thus, knowledge of and liberty to practice culture and spirituality, including the harvesting and use of traditional medicines and healing approaches to support wellbeing and non-medical needs of patients are crucial [28]. Reconnection to lands has been identified as a determinant of health [80], with capacity to heal the effects of dispossession [81,82]. Higher income and education are understood as strongly correlated with higher self-reported health among Indigenous populations [83]. Therefore, FN communities determining and operating economic ventures aimed at closing income gaps and improving the overall wellbeing of individuals and the community at large, is a significant strength. Tackling abject poverty through education and economic enhancements to empower people and create self-reliance holds promise of resulting in better health [84]. The FN communities in this study recognize this and have devised ways to promote education and entrepreneurship as a way out of poverty. Economic activities create self-sufficiency and opportunities for community members who, despite their qualifications, sometimes do not obtain jobs in urban centres.

## 5. Strengths and Limitations of the Study

The strengths of this study lie in the authentic partnership that saw FN control of the study process—design, data collection, and validation. Therefore, the results/strengths that have been outlined are the direct voices and experiences of the participating FN. However, the themes that emerged are the combined experiences of the Nations who participated. Any extrapolations or applications to other FN or Indigenous communities should be done with care to understand and acknowledge any differences that may exist. The FN who participated in this study do not speak on behalf of all First Nations or Indigenous communities, even where similarities exist. The data collected for this study were at the time of publication more than six years old. Though still relevant, much more is written about and discussed now in terms of decolonizing institutions, particularly health and specifically, mental health. First Nation communities are delving more deeply into traditional medicine and preventative approaches outside of biomedicine. With the current Canadian healthcare system crisis, where accessibility of services to all Canadians is increasingly becoming an issue, health promotion, preventative care, including self-care, and traditional and complementary medicine are areas to be taken seriously within healthcare. The focus of this study was to identify First Nation strengths in primary healthcare. What the respondents discussed was a uniquely Manitoba First Nation approach to primary healthcare and health promotion based in traditional values and ways of knowing that were self-determined within communities.

## 6. Conclusions

The value of FN taking control of healthcare management and implementing self-determined innovations in accordance with specific needs within community-based cultural, historical, and geographical contexts cannot be overstated. FN peoples have highlighted specific areas in which local stamina is expended to achieve holistic health. In addition to advocating for strengthening social determinants of health and fostering preventative measures, including improving access to economic opportunities, many respondents are happy their communities have ways to reduce reliance on external (capital and human) resources and are securing continuity by orienting a collaborative and compassionate community-based workforce. Building partnerships with FN communities to support these efforts at transformation is critical to improving community-based primary healthcare in Manitoba. This building on existing strengths with straightforward direction and towards elevating health and ending precarious health and healthcare in First Nation communities is the call of the FN of this study. Allowing greater control of healthcare programming to the First Nations will provide space for a transformative approach that builds upon definitions of healthcare outside of the biomedical model. This new model is inclusive of traditional, spiritual, and land-based connection.

## Figures and Tables

**Figure 1 ijerph-19-13532-f001:**
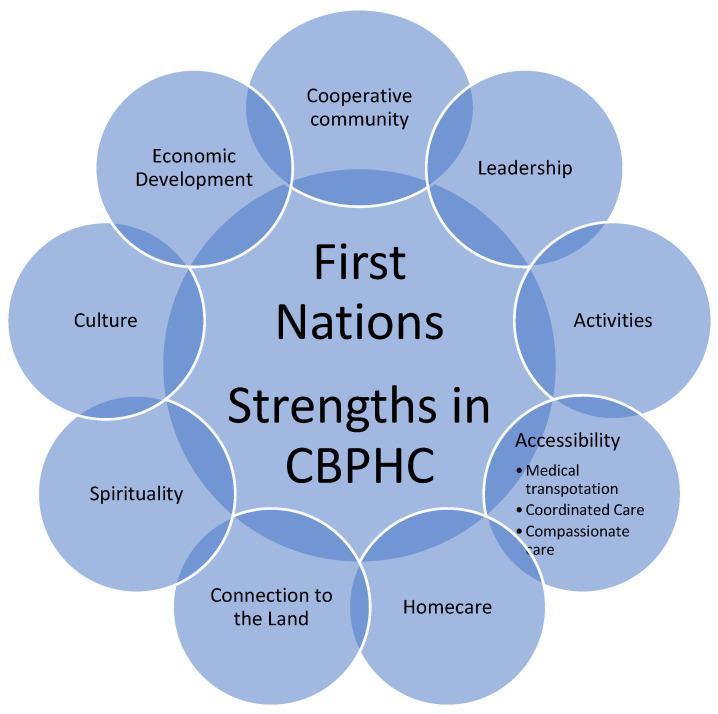
Summary of Findings.

## Data Availability

Data for this study is available upon request by the primary author and is held at First Nation Health & Social Secretariat Research Office.

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
