# Peer review of "Canada First Nations Strengths in Community-Based Primary Healthcare"

_ijerph, 2022, doi:10.3390/ijerph192013532_

Round 1

Reviewer 1 Report

Dear authors, thank you very much for giving me the opportunity to read your manuscript. Because I am trained as a social and cultural anthropologist, I have enjoyed evaluating your research. Your contribution to the field of primary care in indigenous communities in Canada is commendable. Congratulations on a job well done.

However, I would like to make a few suggestions that I think might help to improve the final article.

Introduction:

I think it should be complemented by a background that includes the following topics (which should also be considered in the discussion section):

*cultural colonization

*symbolic efficacy of indigenous peoples' healing practices (Levi Strauss)

*Cultural practices on community health.

*Culturally bounded syndromes (folk diseases) in the context of First Nations research in Canada.

*The importance of self-care outside of the biomedical model (Eduardo Menéndez).

*These and other theoretical elements from medical anthropology.

On the other hand, the journal's rules indicate that the citation model should follow Vancouver standards, but I understand that as this is qualitative research you have followed the APA model.

Method:

It is well written and structured; however, it should follow the COREQ recommendations (COREQ (COnsolidated criteria for REporting Qualitative research)). In fact, you should attach the COREQ checklist to the article as supplementary material (you can easily find it in Tong A, Sainsbury P, Craig J. Consolidated criteria for reporting qualitative research (COREQ): a 32-item checklist for interviews and focus groups. International Journal for Quality in Health Care. 2007. Volume 19, Number 6: pp. 349 - 357).

You have used Grounded Theory as a qualitative approach to analysis. You should cite the primary source in the text (Glaser, B. G., & Strauss, A. L. (2017). The discovery of grounded theory: Strategies for qualitative research. Routledge.) as it was Glaser and Strauss who developed grounded theory. Although Grounded Theory is a good approach. However, and this is not a limitation but only a suggestion, I would have chosen a phenomenological approach to delve deeper into the opinions and experiences of the participants.

Another aspect to highlight is that the data obtained are at least 6 years old. I wonder if nothing has changed between then and now. Perhaps that aspect can be added to the limitations of the study.

The results have been reported according to emerging categories or dimensions. But I wonder if you have not identified sub-themes or sub-categories in the process of coding and qualitative analysis, which is usually an inductive process. Perhaps a greater level of depth in the content analysis of the verbatim interview texts would help the authors to develop a theory from them, which is the purpose of the Grounded Theory analysis on which you rely.

The inclusion of verbatims makes the presentation of the results much clearer. Congratulations, you may have left a lot of fragments out, which is often a source of discouragement for authors of qualitative articles. However, please note that there is no word limit in this journal, which may give you more freedom to expand sections such as the introduction/background.

In the figure showing the categories/themes of the study, it would be very graphic if the different dimensions had different sizes depending on the importance of the dimensions for the participants.

Discussion

I think it is very good that you have highlighted the findings on the strengths of health care in Canada's First Nations, which gives rise to a discourse on the active role of First Nations in health management.

The classic Cultural Anthropology approaches to indigenous health models have always focused on the difficult access to health resources and how this, in part, is due to the different worldviews on illness and health, sometimes very different from those proposed by the hegemonic biomedical model.

However, your study does not question the biomedical paradigm but assumes that it is the valid one, the standard to which indigenous peoples must conform in terms of community health care. I think that a little debate on this would improve your article, which is in itself very valuable and of high quality.

Congratulations and I hope you have good luck with the review.

Author Response

We found the reviewer's comments very helpful and have added the following suggestions. To the introduction, background information of concepts from Medical Anthropology listed by the reviewer below. To the methods section, we have added Glasser & Strauss, 2017 and have appended COREQ. The literature review is reformatted appropriately. Sub-themes are added to themes where possible. The introduction and discussion now have additional information emphasizing First Nation traditions and values re health and healthcare, which emanate from outside biomedicine. 

Thank you so very much for your review.

Rachel

Reviewer 2 Report

Thank you for the opportunity to review this manuscript. Please see the attached document for my comments.

Author Response

Thank you so much for your review. We have gone through each of your suggestions and have added information to the body of the paper in each section. Particularly, community-based methods and GT theory analysis are more eloquently tied together. The sections are more seamlessly and meaningfully connected. The quotes include sub-themes (where we could do) and are contextualized. Information is added to the abstract, introduction, methods, discussion and study limitations.  

Rachel

Reviewer 3 Report

This is an excellent paper. Its strength lies in its First Nations-centric approach to the delivery of primary healthcare. Shockingly poor health statistics is but one of the many deeply problematic effects of British colonisation that has wrought havoc in Indigenous communities throughout the world. Indigenous Peoples have fought for centuries for their right to self-determination and this paper demonstrates clearly the benefits that can accrue to those communities who are sufficiently freed from colonial oppression to focus on their own strengths and to enhance them. Returning decision-making and power to Indigenous communities has always been the key to overcoming the negative effects of colonialism. This paper describes clearly, logically and in considerable detail that is wholly relevant to and respectful of the First Nations communities involved, how to achieve that in community-based health research. It is a very important contribution to the literature.

Author Response

Thank you so very much for your review.

Rachel